# Sequential Application of Different Types of Coagulants as an Innovative Method of Phosphorus Inactivation, on the Example of Lake Mielenko, Poland

Jolanta Katarzyna Grochowska *[iD], Michał Łopata [iD], Renata Augustyniak-Tunowska [iD] and Renata Tandyrak [iD]

Department of Water Protection Engineering and Environmental Microbiology, Faculty of Geoengineering, University of Warmia and Mazury in Olsztyn, St. Prawocheńskiego 1, 10-720 Olsztyn, Poland; michal.lopata@uwm.edu.pl (M.Ł.); rbrzoza@uwm.edu.pl (R.A.-T.); renatat@uwm.edu.pl (R.T.)
* Correspondence: jgroch@uwm.edu.pl

**Abstract:** The process of accelerated eutrophication forces the search for innovative, effective methods to restore the quality of surface waters. This study was conducted on shallow, urban Lake Mielenko (Maximum depth 1.9 m; Mean depth 1.3 m) in the context of implementing a new, sustainable method of lake restoration, i.e., phosphorus inactivation by sequential application of two types of coagulants. Approximately 9.9 tons of polyaluminium chloride (trade name PAX 18) were introduced into the profundal zone of Lake Mielenko, and 9.0 tons of iron chloride (trade name PIX 111) in the coastal area. The applications were divided into two spring and two autumn stages. Before restoration, the mean $P_{min.}$ concentration in Lake Mielenko water was 0.031 mg P/L, and TP was in the range of 0.091 to 0.346 mg P/L. After restoration, the average content of $P_{min.}$ was 0.007 mg P/L (a decrease of 80%), and the average value of TP was 0.096 mg P/L (a decrease of 72%). The obtained results indicate that phosphorus inactivation does not change nitrogen compounds' content. However, due to the application of coagulants, P content decreased, i.e., the main factor limiting photosynthesis, which resulted in a significant decrease in primary production in Lake Mielenko. Before restoration, the average content of $N-NH_4$ was 0.100 mg N/L, $N-NO_3$—0.145 mg N/L, $N_{org.}$ 1.70 mg N/L, and TN—1.86 mg N/L. After restoration, the average content of $N-NH_4$ was 0.096 mg N/L, $N-NO_3$—0.123 mg N/L, $N_{org.}$ 1.28 mg N/L, and TN—1.50 mg N/L. This also resulted in a reduction in the $N_{org.}$ content, as well as a reduction in the amount of chlorophyll a (from 30.51 to 13.41 mg/m$^3$), organic compounds ($BOD_5$ from 8.9 to 4.6 mg $O_2$/L), and an increase in water transparency to the bottom (up to 1.45 m on average). The results obtained in Lake Mielenko indicate that the innovative method of phosphorus inactivation, which involves the sequential application of two types of phosphorus-binding preparations, is an excellent solution that ensures higher ecological safety in the coastal areas of the reservoir and also allows for a significant reduction in restoration costs.

**Keywords:** restoration of lake; P; N; chlorophyll a; Secchi disc visibility

## 1. Introduction

The accelerated process of eutrophication is one of the general, global problems for inland water. This process is caused by anthropogenic activities, mainly urbanization, industrialization, the increase in the area of arable land around lakes, and the clearing of forests [1–3]. Urban lakes, which serve as sewage receivers, are severely affected by the issue of increased eutrophication. According to many researchers [4–9], excessive loading of water with nutrients flowing from the transformed catchments (external load) causes many unfavorable phenomena. The most noticeable problems are: a decrease in water transparency, loss of biodiversity, the proliferation of troublesome species of algae and blue-green algae, a decrease in the visual qualities of water, odors associated with the formation of products of putrefaction processes, and the impairment of habitats of valuable fish species, such as salmonids. The most important symptom of lake pollution is the

activation of the process of internal loading of P and N from bottom sediment [10], which is the effect of increased photosynthesis and leads to disturbances in $O_2$ settings manifested by over-oxygenation of surface water layers and deoxygenation of bottom layers, and sometimes even complete anoxia of the aquatic environment (oxygen consumption for organic decomposition processes) [11,12]. The anoxic condition in the water above the sediment causes a decrease in the redox potential and the start of internal loading. Bottom sediments become an endless source of phosphorus, which is the main element determining the fertility of water and the size and intensity of production processes in water bodies [13]. Because water is a raw material of fundamental importance for the living conditions and economic development of the world, the accelerated eutrophication of water bodies forces the search for appropriate methods that will reverse, stop, or at least slow down this process and its adverse consequences. Also, the provisions of the Water Framework Directive [14] oblige all Member States to take measures to protect and sustain the use of inland surface waters. To improve the quality of lake waters, various protective measures are taken, consisting of limiting the inflow of biogenic substances from external sources, i.e., cutting off point sources (municipal and industrial sewage), improving water and sewage management in communes, introducing rational agricultural management, and using phosphorus-absorbing aggregates in the beds of watercourses feeding lakes [15–20]. However, these actions can bring good results only in the case of poorly or moderately eutrophicated lakes. The lack of reaction of degraded lakes to the reduction of the external load is most often caused by the phenomenon of internal loading. In this case, it is necessary to carry out restoration procedures [21]. Over the last 60 years, a number of different remediation treatments have been tried around the world, such as hypolimnetic withdrawal, flushing-dilution, artificial mixing, hypolimnetic aeration, bottom sediment dredging, bottom sediment treatment, and different types of biomanipulation [22–32].

One of the most commonly used methods of lake restoration is phosphorus inactivation, consisting of introducing preparations into the waters that bind phosphates and immobilize them in bottom sediment [21]. Iron, aluminum, or calcium salts are most often used for phosphorus inactivation [33,34]. Iron and aluminum coagulants are mainly chlorides and sulfates dosed in water in the form of acidic solutions [21,22]. In contact with lake water, these salts undergo hydrolysis, and the resulting metal hydroxides precipitate, forming flocs that coagulate the suspension from the water column and fall to the sediment surface, creating a microlayer rich in metals that bind phosphorus and a barrier for phosphates released from bottom sediment [33,34].

An innovative, sustainable solution developed by our scientific team is the restoration of lakes via the modified P inactivation method. This research aims to implement a new method based on the sequential application of coagulants, iron chloride (trade name PIX 111) and polyaluminium chloride (trade name PAX 18), and to determine the effectiveness of this innovative method in the restoration of a shallow, polymictic urban lake. The Fe coagulant was dosed in the coastal zone of the lake, where the water is well-oxygenated. The aluminum coagulant was dosed in the profundal zone, where the deep, over-bottom water is anoxic. This solution enables the reduction of restoration costs and ensures a higher level of ecological safety.

## 2. Methodology

### 2.1. Description of This Study Area

Lake Mielenko (54°19′55″ N, 18°10′55″ E, 204.0 m AMSL) is located in the city of Kartuzy, in the macroregion of East Pomeranian Lakeland, and in the mesoregion of Kashubian Lakeland, Poland (Figure 1) [35]. The climate of this region is cool and humid, with a short growing season that lasts 152 days, from May to October. In this period, the average daily temperature exceeds 17 °C, which ensures the optimal development of plants and animals. In Kartuzy, the mean annual air temperature is 6.5 °C. The hottest month of the year is July, when the mean daily temperature is 21 °C, and the coldest is January, when the average daily temperature is −5 °C. The average annual rainfall in this

region is 750 mm. The month with the highest rainfall is July—the average daily rainfall is 59 mm. The month with the least rainfall is February—the mean daily rainfall is 13 mm. The period of the year with snowfall lasts 5 months—from November to April—and the average monthly snowfall during this time is 25 mm. The snowiest month is December, where the average monthly snowfall is 109 mm.

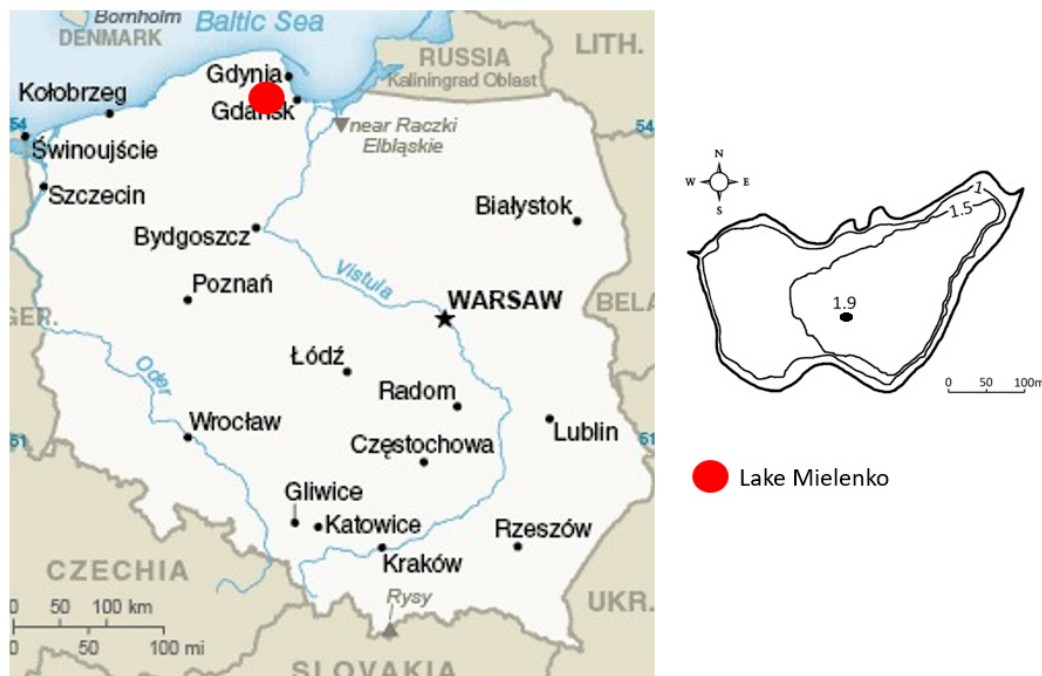

**Figure 1.** The situation map of Lake Mielenko in Poland (https://upload.wikimedia.org/wikipedia/commons/1/17/Poland-CIA_WFB_Map.png, accessed on 30 August 2023).

The surface area of the lake is small and amounts to 7.8 ha. The maximum depth of this study water body is 1.9 m, and the mean depth is 1.3 m. The lake retains 102,900 $m^3$ of water. The length of Lake Mielenko is 460 m, and the width is 252 m. The 1314 m long shoreline is poorly diversified because its development index is 1.3. Lake Mielenko is used for recreation (fishing) by the inhabitants of Kartuzy.

A stream flows into Lake Mielenko, draining the peat bog areas. The outflow is towards Lake Karczemne through a stream called Klasztorna Struga. The total catchment area of Lake Mielenko is 3.82 $km^2$, and the direct catchment area is 0.22 $km^2$. The method of developing these areas is shown in Figure 2.

The acceptable and dangerous loads of phosphorus for Lake Mielenko calculated from the hydrological model of Vollenweider [36] are, respectively, 0.050 g P $m^{-2}$ $year^{-1}$ and 0.100 mg P $m^{-2}$ $year^{-1}$ (3.9 and 7.8 kg P $year^{-1}$).

During the research conducted in the years 2018–2022, it was found that Lake Mielenko is supplied with biogenic salts from area sources, i.e., surface runoff from the direct catchment area, atmospheric sources, and the inflow of rainwater from the shore outlet. The average P load introduced annually into Lake Mielenko is 0.081 g P $m^{-2}$ $year^{-1}$ (6.3 kg P $year^{-1}$).

A comparison of the actual phosphorus load on the lake with the calculated loads from the model of Vollenweider [36] indicates that the allowable load was exceeded 1.5 times, but the dangerous load causing avalanche eutrophication was not exceeded.

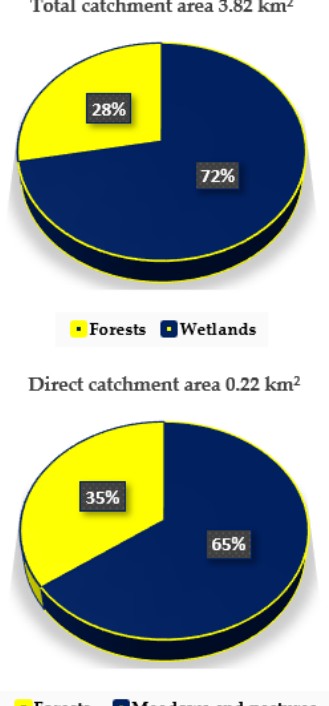

**Figure 2.** The use of the total and direct catchment area of Lake Mielenko.

## 2.2. The Collection of Water Samples and Laboratory Analysis

Water samples from Lake Mielenko were collected monthly from January to December 2018, 2019, 2020, 2021, and 2022. According to the guidelines used in limnological research, water samples for analysis should be collected at the deepest point of the lake (1.9 m, location according to bathymetric maps and GPS). Due to the small depth of Lake Mielenko, samples were taken from a depth of 1 m, but at three points located at the vertices of a triangle with sides of 50 m. These points were located in the area of the maximum depth of the lake. During this research, a total of 180 water samples were taken for analysis. Water for analysis was collected using a Ruttner sampler (3.5 L, KC Denmark; Geomor Technik, Szczecin, Poland) into 2 L containers.

Water temperature (T), oxygen content ($O_2$), EC, and chlorophyll a were measured using a YSI 6600V2 multiparameter sensor (Yellow Springs, OH, USA).

Measurements of water transparency (SD) were made with a Secchi disk (ø 0.30 m, KC Denmark, Geomor Technik, Szczecin, Poland).

Phosphorus forms ($P_{min.}$ and TP) and nitrogen forms (ammonium nitrogen $N-NH_4$, nitrate nitrogen $N-NO_3$, TN) were analyzed according to the Standard Methods [37]. Organic forms of nutrients were calculated as the difference between total and mineral amounts of nutrients. TOC was measured using a TOC-TN nitrogen and carbon analyzer by Hach, Loveland, Colorado, USA. Seston was analyzed by the gravimetric method using a glass fiber filter called Whatman GF/C, and $BOD_5$ was determined by the direct method [37].

Analyses of water samples were performed at the Laboratory of the Department of Water Protection Engineering and Environmental Microbiology of the University of Warmia and Mazury in Olsztyn, Poland.

Every analysis was made in triplication. The coefficient of variation (CV) for the repeated analysis was 2% [38].

The results of $P_{min.}$, $P_{org.}$, TP, $N-NH_4$, $N-NO_3$, $N_{org.}$, TN, chlorophyll a, and SD were statistically analyzed (one-way ANOVA, *p* = 0.05, Tukey's HSD) using a Statistica 13.3 software package [39]. An alternative hypothesis tested was the presence of significant differences in mean annual values of tested parameters between 2018 (protective actions)

and 2019 (after protective actions), 2020, 2021 (during restoration treatments), and 2022 (after restoration treatments). Results of TOC, $BOD_5$, seston, pH, and EC were presented in the form of a table containing annual means, standard deviation, and minimum and maximum values.

*2.3. Restoration of Lake Mielenko*

Comprehensive studies of Lake Mielenko made it possible to adapt the restoration method to the environmental conditions prevailing in the water body. The optimal solution was the method of P inactivation using two types of coagulants, i.e., Fe and Al, applied sequentially: iron coagulant in the littoral zone and aluminum in the profundal zone. A number of commercially available and commonly used iron and aluminum coagulants were tested in laboratory conditions, and based on the obtained results of their effectiveness in water purification, PIX 111 (the trade name of the coagulant ferric chloride) and PAX 18 (the trade name of polyaluminium chloride) were selected.

The application of coagulants was divided into four stages: spring 2020 and 2021, and autumn 2020 and 2021. Based on the results of analyses of the P fraction content in the bottom sediments of Lake Mielenko, it was calculated that approx. 19 g of $Al/m^2$ is needed to bind P in the profundal sediments of the lake, and for coastal sediments predisposed to the use of iron coagulant, the demand for Fe is 29 g $Fe/m^2$.

With an area of 3.8 ha corresponding to the area of bottom sediments within deoxygenated zones during periods of water stagnation (profundal sediments, the central part of the lake), and with the demand of these sediments for the necessary amount of phosphorus binding agent (19.3 g $Al/m^2$ of sediment), the amount of aluminum was 734.7 kg Al. The demand for binding excess P from the water depths of the lake was calculated on the basis of the optimal Al/P ratio. With the volume of the central zone equal to 62,700 $m^3$, the dose of aluminum for Lake Mielenko was determined to be 159.9 kg Al. This gives a total value of 894.6 kg (for PAX 18 (9 $\pm$ 0.02% Al), the dose is 9940 kg of commercial product).

With an area of 4.0 ha corresponding to the area of littoral (coastal) sediments within the well-oxygenated zone and with the demand of these sediments for the necessary amount of phosphorus binding agent (29.0 g $Fe/m^2$ of sediment), the amount of iron was 1160.1 kg Fe. The demand for the binding of excess phosphorus from the water column of the lake was calculated on the basis of the optimal Fe/P ratio with the volume of the coastal zone equal to 40,200 $m^3$. The dose of iron for Lake Mielenko was determined at 51.3 kg Fe. This gives a total value of 1211.4 kg (for PIX 111 (13.4 $\pm$ 0.6% Fe), the dose is 9040 kg of commercial product).

Reagents were distributed using the surface method from the deck of vessels. The chemical dosing system made it possible to regulate the intensity of their administration.

During application, attempts were made to distribute the coagulants as evenly as possible over the entire surface of the designated water area and to strictly avoid uncontrolled discharges of the agent used during the standstill or maneuvers of auxiliary vessels. The coagulant was fed just below the water surface using technical solutions that prevented the formation of flocs and their flotation.

The doses of coagulants were selected and applied in such a period (spring or autumn) that they did not cause any negative changes in water chemistry and, at the same time, effectively eliminated the mineral phosphorus content in the water. It should be clearly emphasized that all coagulant components are elements commonly found in nature. During the application of coagulant, iron concentrations in the water of Lake Mielenko did not exceed 0.5 mg Fe/L (the permissible concentration in water is 1 mg Fe/L), and aluminum concentrations during coagulant dosing were traces (the permissible concentration in water is 0.3 mg Al/L).

## 3. Results

It was discovered that there were high differences in the concentrations of $P_{min}$ between the analyzed research years, as shown in the statistical analysis ($\alpha < 0.02$; F—10.35).

($\alpha < 0.02$; F—10.35). In 2018 and 2019, before restoration treatments, the average $P_{min.}$ was $0.031 \pm 0.013$ mg P/L. In 2020 (during the first stage of phosphorus inactivation), the average concentration of $P_{min.}$ decreased to $0.016 \pm 0.008$ mg P/L (Figure 3). In the following year, when coagulants were applied, a further reduction in the content of $P_{min.}$ in the lake water was achieved—an average value of $0.012 \pm 0.014$ mg P/L. In 2022, the water of the research lake contains the lowest content of $P_{min.}$—on average, $0.007 \pm 0.007$ mg P/L (Figure 3).

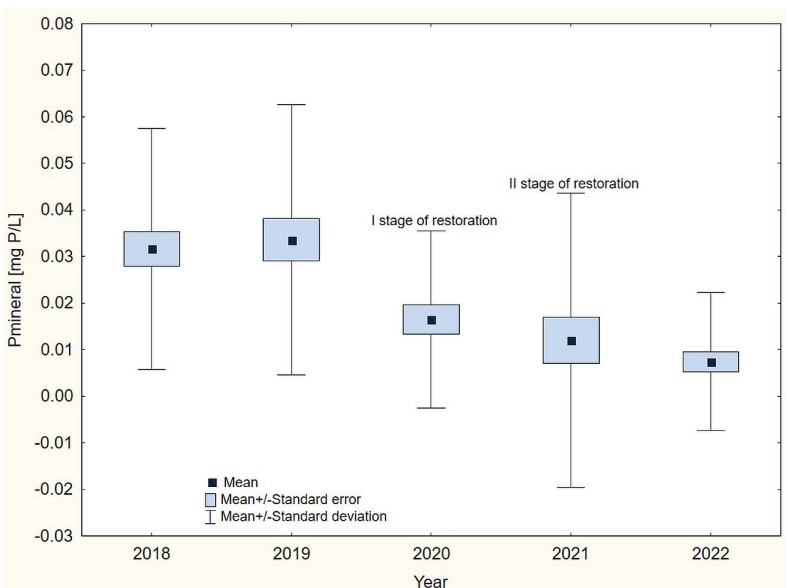

**Figure 3.** The mean values of $P_{min.}$ in the water of this study body.

It was discovered that there were high differences in the concentrations of $P_{org.}$ between the analyzed research years, as shown in the statistical analysis ($\alpha$—0.107; F—1.99). Before restoration treatments, the average content of $P_{org.}$ was $0.153 \pm 0.083$ mg P/L and after the end of phosphorus inactivation treatments (2022), it was $0.088 \pm 0.029$ mg P/L (Figure 4).

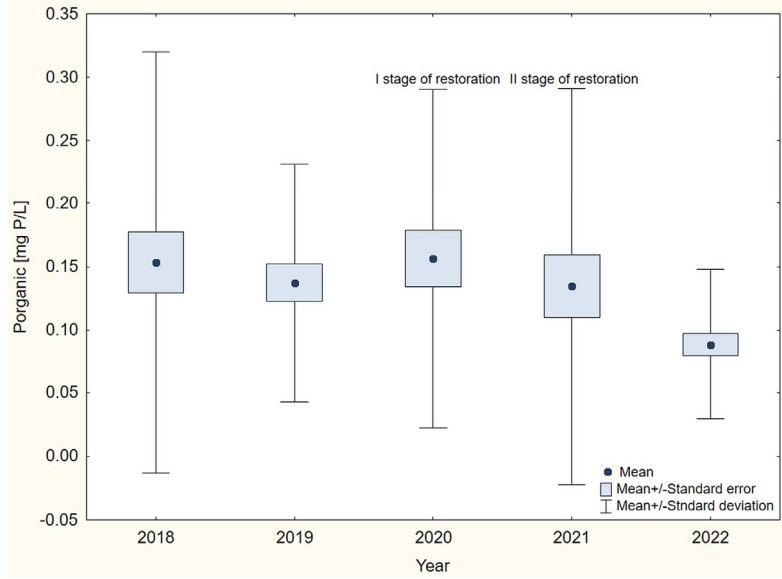

**Figure 4.** The mean annual values of $P_{org.}$ in the water of this study body.

It was discovered that there were high differences in the concentrations of TP between the analyzed research years, as shown in the statistical analysis ($\alpha$—0.064; F—2.36). In effect of restoration treatments, the mean values of TP decreased from $0.185 \pm 0.083$ mg P/L (in 2018) to $0.096 \pm 0.034$ mg P/L (in 2022) (Figure 5).

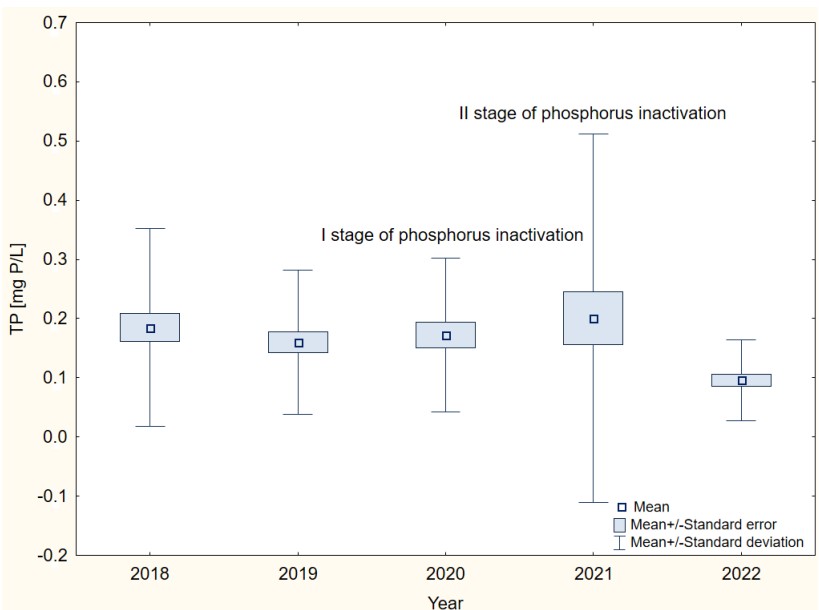

**Figure 5.** The mean annual values of TP in the water of this study body.

It was not discovered that there were high differences in the concentrations of ammonium (N-NH$_4$) between all research years, as shown in the statistical analysis ($\alpha$—0.642; F—0.62). The average amount of N-NH$_4$ in 2018 was $0.101 \pm 0.104$ mg N/L, and after the end of restoration, it was $0.096 \pm 0.086$ mg N/L (Figure 6).

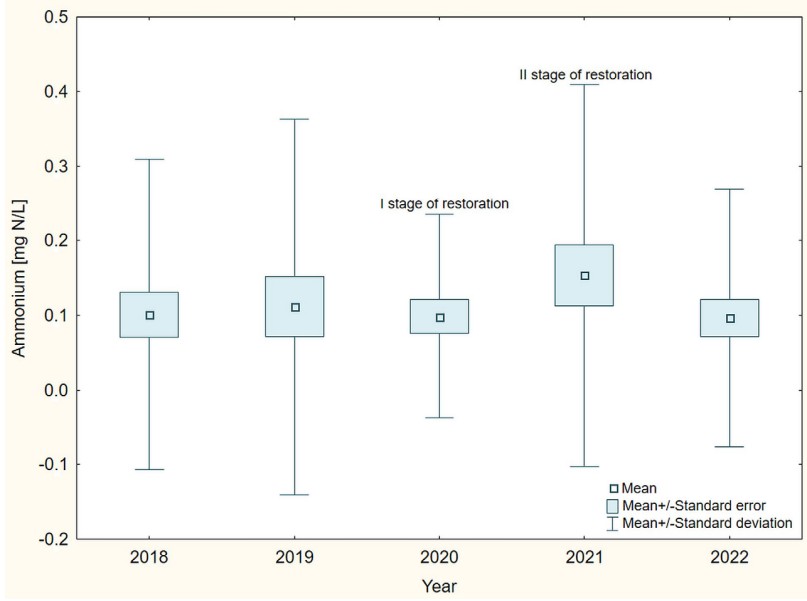

**Figure 6.** The mean annual values of ammonium (N-NH$_4$) in the water of this study body.

It was discovered that there were high differences in the concentrations of nitrate nitrogen (N-NO$_3$) between the analyzed research years, as shown in the statistical analysis ($\alpha < 0.02$; F—11.85). In 2018, the average amount of N-NO$_3$ was $0.145 \pm 0.062$ mg N/L;

in 2019, the mean value of N-NO$_3$ was 0.206 ± 0.057 mg N/L. During the first stage of restoration, the average content of nitrates was 0.132 ± 0.039 mg N/L, and in the second stage of coagulant application, it was 0.076 ± 0.036 mg N/L (Figure 7). In 2022, an increase in the content of N-NO$_3$ to 0.124 ± 0.030 mg N/L was found.

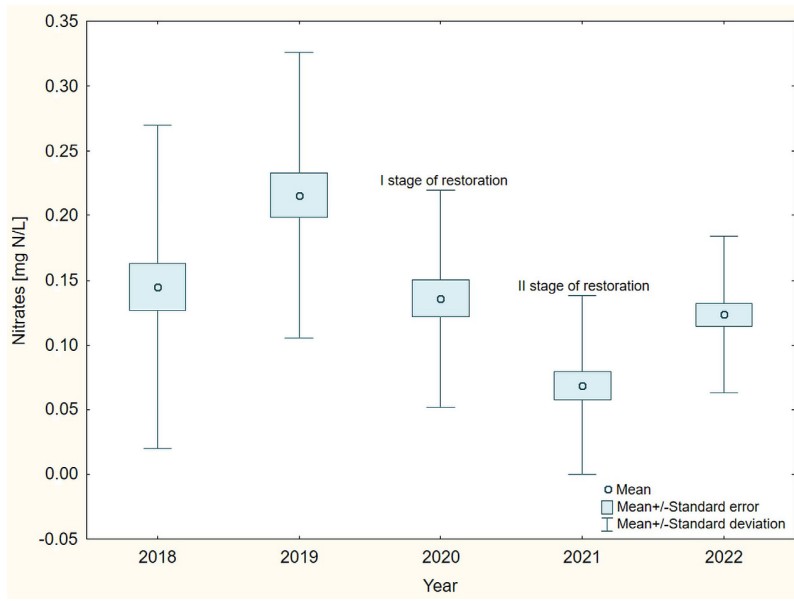

**Figure 7.** The mean annual values of nitrate nitrogen (N-NO$_3$) in the water of this study body.

It was discovered that there were high differences in the concentrations of organic nitrogen (N$_{org.}$) between the analyzed research years, as shown in the statistical analysis (α < 0.02; F—4.62). In 2018 (before restoration), the average content of N$_{org}$ was 1.697 ± 0.605 mg N/L, and in 2022 it was 1.284 ± 0.280 mg N/L (Figure 8).

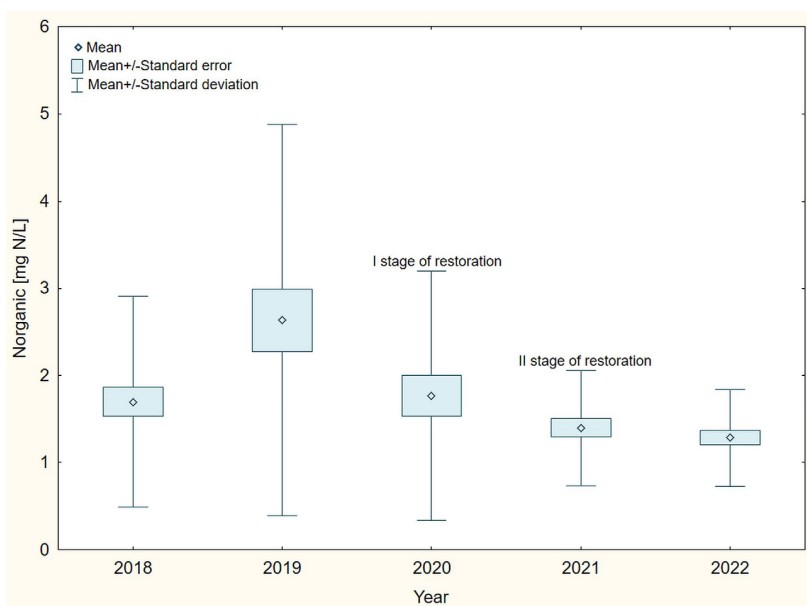

**Figure 8.** The mean annual values of organic nitrogen (N$_{org.}$) in the water of this study body.

It was discovered that there were high differences in the concentrations of organic nitrogen (N$_{org.}$) between the analyzed research years, as shown in the statistical analysis (α < 0.02; F—6.45). In 2019, the mean value of total nitrogen was 2.723 ± 1.134 mg N/L. When the P inactivation measures were carried out on the lake (2020 and 2021), the average

concentration of TN ranged from $1.633 \pm 0.635$ to $1.493 \pm 0.392$ mg N/L (Figure 8). In 2022, a slight decrease in TN was noted—$1.284 \pm 0.280$ mg N/L (Figure 9).

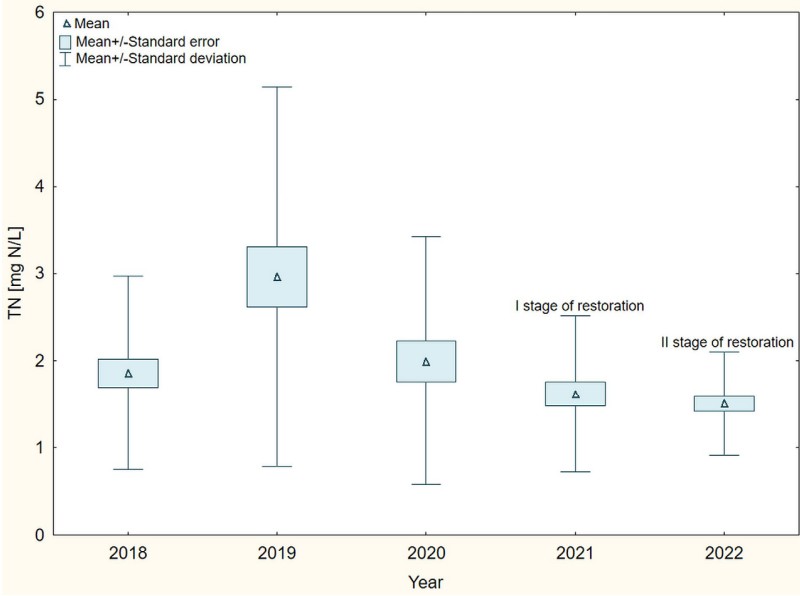

**Figure 9.** The mean annual values of total nitrogen (TN) in the water of this study body.

In the water of this study lake in the control year (2018), the mean amount of chlorophyll a was $30.51 \pm 11.13$ mg/m$^3$. During P inactivation, the mean values of chlorophyll a ranged between $15.44 \pm 13.56$ and $16.38 \pm 8.44$ mg/m$^3$ (Figure 10). After the end of restoration treatments (2022), a further decrease in the chlorophyll a content to mean values of $13.41 \pm 9.49$ mg/m$^3$ was noted (Figure 10).

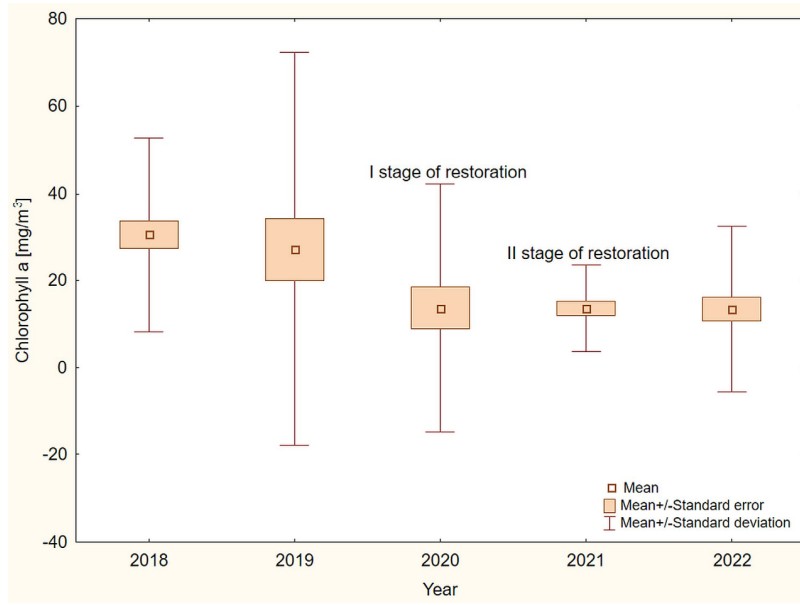

**Figure 10.** The mean annual values of chlorophyll a in the water of this study body.

Before the restoration measures (2018), the average value of water transparency was 0.4 m (Figure 11). The P inactivation method influenced the improvement of Secchi disc visibility. The average level of water transparency was 1.2 m in the second stage of the treatment. In 2022, after the end of restoration procedures, the mean value of this parameter reached 1.45 m (Figure 11).

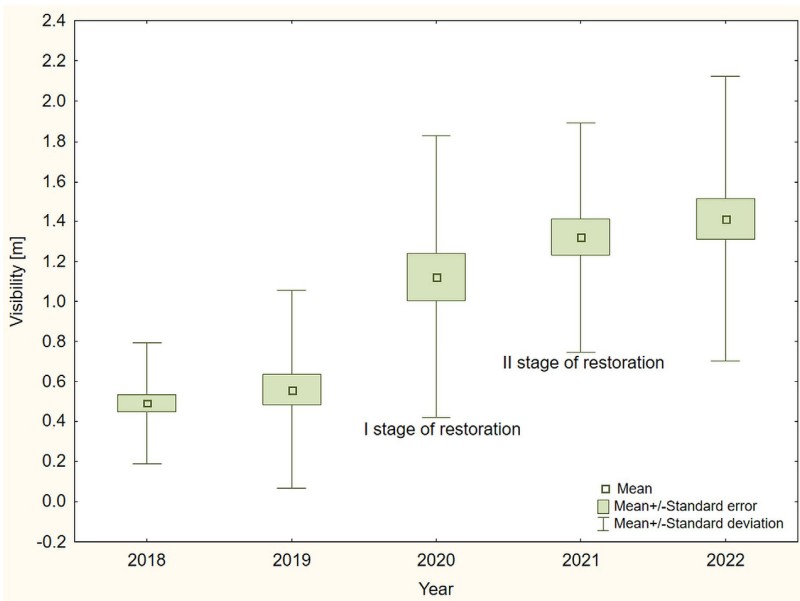

**Figure 11.** The mean annual values of Secchi disc (SD) visibility in study water body.

A comparison of other water parameters Lake Mielenko before and after restoration showed significant changes in their values. The average content of hardly degradable organic matter expressed as TOC (total organic carbon) decreased from $26.97 \pm 4.19$ mg C/L to $14.91 \pm 1.89$ mg C/L, and the parameter specifying the amount of autochthonous organic matter, $BZT_5$, decreased from $8.9 \pm 1.9$ mg $O_2$/L up to $4.6 \pm 1.6$ mg $O_2$/L (Table 1). A reduction in the amount of suspended solids from the average value of $10.2 \pm 5.4$ mg/L to $5.0 \pm 2.5$ mg/L was also noted (Table 1).

**Table 1.** The annual variability of the selected physicochemical indicators of Lake Mielenko.

|  |  | TOC [mg/L] | BOD₅ [mg/L] | Reaction [pH] | EC [µS/cm] | Seston [mg/L] |
|---|---|---|---|---|---|---|
| 2018 | Average | 26.97 | 8.9 | 8.15 | 419 | 10.2 |
|  | Standard Deviation | 4.19 | 1.9 | 0.54 | 29 | 5.4 |
|  | Minimum | 21.55 | 7.0 | 7.55 | 382 | 4.3 |
|  | Maximum | 33.14 | 12.5 | 9.14 | 475 | 20.8 |
| 2019 | Average | 18.51 | 6.0 | 8.48 | 660 | 13.7 |
|  | Standard Deviation | 4.86 | 1.9 | 0.52 | 92 | 8.8 |
|  | Minimum | 12.95 | 3.4 | 7.50 | 566 | 2.0 |
|  | Maximum | 28.5 | 10.1 | 9.81 | 793 | 33.0 |
| 2020 | Average | 17.91 | 5.4 | 7.83 | 689 | 8.0 |
|  | Standard Deviation | 7.88 | 1.1 | 0.51 | 47 | 3.9 |
|  | Minimum | 11.01 | 3.9 | 6.71 | 610 | 1.4 |
|  | Maximum | 35.43 | 7.3 | 8.50 | 737 | 14.5 |
| 2021 | Average | 13.33 | 5.2 | 7.97 | 754 | 5.4 |
|  | Standard Deviation | 1.30 | 2.0 | 0.24 | 27 | 2.6 |
|  | Minimum | 11.30 | 1.6 | 7.67 | 700 | 0.8 |
|  | Maximum | 16.16 | 8.9 | 8.43 | 794 | 11.0 |
| 2022 | Average | 14.91 | 4.6 | 8.13 | 647 | 5.0 |
|  | Standard Deviation | 1.89 | 1.6 | 0.40 | 77 | 2.5 |
|  | Minimum | 11.51 | 2.4 | 7.51 | 517 | 1.6 |
|  | Maximum | 18.24 | 7.5 | 8.78 | 736 | 9.2 |

## 4. Discussion

The most common types of lakes in the world are shallow water bodies [40], and due to unfavorable morphometric conditions and high water dynamics, they are particularly susceptible to eutrophication, leading to an increase in water fertility and, as a result, to their rapid disappearance [41,42]. The term "shallow" lake refers to water bodies that are less than 10 m deep and do not undergo permanent thermal stratification [43]. Lake Mielenko is classified as a "shallow" lake because its maximum depth is only 1.9 m, and it is also a polymictic lake. During research conducted in 2018–2022, the presence of thermal stratification was never detected, except during the period of ice. The water temperature was practically the same at the surface and in the bottom layers, and it changed throughout the year in line with the air temperature.

Lake Mielenko was a recipient of rainwater and water from fishponds and was intensively used by anglers using excessive amounts of bait. The lake is naturally fed by the waters of a small stream that flows through marshy areas. The high load of nutrients flowing from the catchment and the high susceptibility of Lake Mielenko to degradation resulted in its contamination. Protective measures in the reservoir's catchment area, which included cutting off the water supply from breeding facilities and limiting the use of baits, were insufficient to improve the lake's water quality, which indicated the need to use an appropriately selected restoration method.

In the case of shallow lakes, selecting the optimal restoration method is very difficult due to the strong coupling between sediments and water [40,41]. According to literature data [42–44], restoration methods dedicated mainly to shallow, polymictic lakes include flushing, removal of bottom sediments, isolation of bottom sediments, biomanipulation, and phosphorus inactivation. The optimal solution for Lake Mielenko was the phosphorus inactivation method with an innovative solution consisting of the sequential application of two types of coagulants, i.e., iron (PIX 111) and aluminum (PAX 18). As is known, all restoration methods are primarily aimed at reducing the content of phosphorus in water, the most important element responsible for the eutrophication process of lakes, to values limiting primary production [45,46]. Chemical transformations of phosphorus in lake water lead to its precipitation and accumulation in bottom sediments in the form of detritus and in the form of connections with Fe, Ca, Mn, Al, and organic matter, until the sorption capacity of bottom sediments is exhausted [47,48]. As research has shown [49], phosphates have the greatest affinity for iron. For this reason, the use of an iron coagulant (PIX 111) was justified to inactivate phosphorus in Lake Mielenko. It should be noted, however, that the direction of binding of phosphorus with iron in bottom sediments is determined by the reaction and, above all, the oxygenation of bottom waters [50,51]. In lakes where the oxygen concentration in the water at the bottom drops to 2 mg $O_2$/L and below that, the redox potential decreases. At a potential of 0.2 V, the reduction of iron from +3 to +2 takes place. When iron is in the 3rd oxidation state, phosphates are permanently combined with it in bottom sediments, while when Fe turns into a reduced form, P-$PO_4$ is released into water [52]. At a value of 0.1 V and lower, sulfates are reduced to hydrogen sulfide. Hydrogen sulfide combines with iron to form sulfides on the surface of bottom sediments [53]. Iron bound in the form of sulfides cannot participate in the precipitation of phosphates. On the other hand, a necessary condition for initiating the process of binding phosphorus with iron in bottom sediments with good water oxygenation is a Fe/P ratio of at least 1.8 and even higher than 3 [54,55]. In Lake Mielenko, the Fe/P ratio in the water ranged from 0.6 to 6.2/1, and on average it was 3.2/1, so the precipitation of phosphorus into bottom sediments was limited. The application of iron coagulant could have resulted in an increase in the Fe/P ratio. However, the iron coagulant could only be used in the coastal zone of Lake Mielenko, where the waters were well oxygenated. The coastal zone is also a habitat for many plant and animal organisms, which is another factor supporting the need to use an iron coagulant in this zone to maintain greater ecological safety.

Despite the high dynamics of the waters of Lake Mielenko, in 2018 and 2019, progressive deoxidation of the bottom waters was recorded in the profundal area. This situation

resulted from the abundance of organic matter in the lake's waters and bottom sediments. The source of organic matter was primarily the production processes taking place within the ecosystem, but also the inflow of water from fishponds, water from marshy areas, and the use of baits. In shallow water bodies, organic material is also quickly deposited in bottom sediments and is therefore exposed to oxidation processes taking place in the water for a shorter time. Organic matter underwent mineralization, which resulted in oxygen depletion at the bottom. This process also occurred at a rapid pace due to the high temperature of the bottom waters (approximately 20 °C), as determined by polymixis. The rate of decomposition of organic matter depends on its chemical composition [56]. Organic matter of algal origin, typical of polluted lakes with high primary production, decomposes quickly because it is made of material with a low C/N ratio. In turn, macrophytes are rich in cellulose; hence, they are characterized by a higher value of this ratio and less susceptibility to decomposition [57]. The water and bottom sediments of Lake Mielenko also contained humic substances flowing from marshy areas. The best effects of purifying water rich in organic substances, including humic substances, are achieved through the use of polyaluminium chloride [58,59]. Additionally, the aluminum coagulant creates permanent connections with phosphorus, even in anaerobic conditions. For this reason, the aluminum coagulant PAX 18 was applied to the profundal zone. Before restoration, the phosphate content in the waters of Lake Mielenko was not very high—from 0.021 to 0.065 mg/L (average 0.031 mg P/L). Total phosphorus at that time ranged from 0.091 to 0.346 mg P/L (average 0.135 mg P/L). The amount of total phosphorus was mainly determined by its organic form. After the introduction of the first and subsequent doses of the coagulant, phosphates were not completely removed from the water, but their concentrations decreased significantly, which resulted in a reduction in primary production. The restoration method used does not directly affect the organic phosphorus content. The visible organic matter reduction was the result of a reduction in production processes due to a radical reduction in the availability of phosphates. The P inactivation method also contributed to reducing the total amount of phosphorus in Lake Mielenko. In 2022, i.e., in the control year after the fourth stage of restoration using this method, the concentrations of total phosphorus were reduced by 50% and ranged from 0.048 to 0.170 mg P/L.

The P inactivation method is a procedure that mainly causes phosphorus to be precipitated from the water column and permanently immobilized in bottom sediments. The obtained results indicate that phosphorus inactivation basically does not change the content of N compounds. However, as a result of the application of coagulants, the total phosphorus content decreased, i.e., the main factor limiting photosynthesis, which resulted in a significant decrease in primary production in Lake Mielenko. This also resulted in a reduction in the organic nitrogen content, as well as a reduction in the amount of chlorophyll a (from 30.51 to 13.41 $\mu$g/L), organic compounds ($BOD_5$ from 8.9 to 4.6 mg $O_2$/L), and an increase in water transparency to the bottom (up to 1.45 m on average). It should be emphasized that the growth of the nitrogen-to-phosphorus ratio (N/P) is the effect of the precipitation of phosphorus from the water column into bottom sediments. This is a very important indicator for determining the dominance of specific algae species [60]. As research has shown, increasing the N/P ratio above 14 causes the disappearance of cyanobacterial blooms and the growth of more ecologically desirable algae species [61]. Before the restoration of Lake Mielenko, during the growing season, the N/P ratio varied from 8 to 11, while after the completion of the reservoir renovation process, it ranged from 15 to 32. Such a change clearly proves the improvement of environmental conditions in the lake, which positively affects the increase in biodiversity.

The innovative method of phosphorus inactivation proposed by our team using two types of coagulants—aluminum and iron—is a very cheap technological solution. The cost of purchasing coagulants with the application is approximately EUR 15,000. The restoration of Lake Mielenko required water law consent, which was issued by the Polish Waters State Water Management Company based on a water law report.

## 5. Conclusions

Research has shown that good practice in lake restoration is the application of two types of coagulants. Both coagulants (PIX 111 and PAX 18) supported each other. The surface distribution of chemical preparations stimulated the reduction of phosphates from the lake water and, first of all, an effective immobilization in the bottom sediment. Improving the chemical properties of lake water and reducing primary production are the first effects of the treatment. If the chemical conditions in the lake are stable, another effect of restoration is changes in the biological elements of the ecosystem, such as phytoplankton, zooplankton, macrophytes, and ichthyofauna. The effects obtained on Lake Mielenko indicate that the described restoration method can be successfully used in shallow, polluted urban lakes. Moreover, the obtained results showed that using Fe and Al salts makes it possible to reduce the restoration cost, and the dosing of iron salts in the coastal areas of the lakes is safer for aquatic organisms in the event of unforeseen changes such as a drop in water pH.

**Author Contributions:** Conceptualization, J.K.G. and M.Ł., investigation J.K.G., M.Ł., R.T. and R.A.-T., verification R.A.-T. and R.T., writing-original draft preparation J.K.G. and R.A.-T., writing—reviewing and editing J.K.G., R.A.-T., M.Ł. and R.T. All authors have read and agreed to the published version of the manuscript.

**Funding:** Project financially supported by the Minister of Education and Science under the program entitled "Regional Initiative of Excellence" for the years 2019–2023, Project No. 010/RID/2018/19, amount of funding 12.000.000 PLN.

**Institutional Review Board Statement:** Not applicable.

**Informed Consent Statement:** Not applicable.

**Data Availability Statement:** Data are available at Department of Water Protection Engineering and Environmental Microbiology.

**Acknowledgments:** The authors thank the community of Kartuzy.

**Conflicts of Interest:** Authors declare no conflict of interest.

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
