# Peer review of "Sequential Application of Different Types of Coagulants as an Innovative Method of Phosphorus Inactivation, on the Example of Lake Mielenko, Poland"

_sustainability, doi:10.3390/su152316346_

Round 1

Reviewer 1 Report

Comments and Suggestions for Authors

In this study, the authors investigated SEQUENTIAL APPLICATION OF DIFFERENT TYPE OF COAGULANTS AS AN INNOVATIVE METHOD OF PHOSPHORUS INACTIVATION”. Firstly, data of parameters have shown in the Abstract. For example,The obtained results indicate that phosphorus inactivation basically does not change the content of nitrogen compounds. Where are the content of nitrogen compounds, how is the content of nitrogen compounds. Secondly, what is implementing a new method of lake restoration of phosphorus inactivation? The title have emphasized the new method, in the context of manuscript has narrated in the Material and methods section. Thirdly, in theMaterial and methods section, where are the statistical analysis ? Thus, I suggest major revisions. 

Comments on the Quality of English Language

Minor editing of English language required

Author Response

Dear Reviewer

We want to thank you very much for the review. We think your comments are very accurate and necessary to improve the quality of the manuscript. We made all your remarks in the text. We hope that the changes introduced by me following your recommendations will be sufficient for the manuscript to be accepted for publication.

  1. Dear Reviewer, an innovative solution in the recultivation method, i.e. phosphorus inactivation in the case of Lake Mielenko, involves using two types of coagulants and their sequential application. First of all, in the coastal areas of the lake where the bottom layers of water are well-oxygenated, an iron coagulant is used. Iron is an element completely safe for aquatic organisms, and iron coagulant is three times cheaper. The remaining part of the lake, where anaerobic conditions occur in the bottom waters, is being restored using an Al coagulant. The aluminum coagulant causes the precipitation of mineral phosphorus from the water column and permanently blocks phosphates in bottom sediments, preventing the release of phosphates from the sediments even in the absence of oxygen.
  2. We have added a range of nitrogen compound changes to the abstract.

Reviewer 2 Report

Comments and Suggestions for Authors

Although the title of the manuscript is attractive and its results can have applications in restoration of lakes, there are many important deficiencies in the manuscript that should be addressed. I have listed them below:

A major concern with this manuscript is its writing style, as some parts are badly written with many sentences being long and boring! In this regard, the manuscript should be revised and rewritten.

The title should be modified to “Sequential Application of Different Type of Coagulants as an Innovative Method of Phosphorous Inactivation, Case Study from Lake Mielenko, Poland”.

Based on what is said in lines 94-101, it is not clear whether the discussed method is proposed by the authors in this study or this method was proposed by others and the authors intend to evaluate its effectiveness! If the former is true, it should be reflected in the entire manuscript and especially in the Materials and Methods section. If the latter is true, haven't the proposers of the method checked its capability?! Specify this.

Lines 104-105: You are writing an article with an international audience, so write the name of the country where this lake is located because not everyone in the world knows that this small lake is located in Poland!

Figure 1 is not of good quality. It is better to provide a larger-scale map (not a Google Earth image) that shows its geographical location in Poland (not in Europe) in relation to its nearby features.

Provide data on climate conditions including rainfall, evaporation, weather type, elevation, etc., as well as the probable local water use of the lake, surrounding population, ecosystem characteristics of the lake, etc. in Section 2.1.

In section 2.2, it should be explained about the criteria used in selecting the number of samples and their location. In fact, how was your sampling network and based on what criteria?!

In section 2.3, it is necessary to explain the method and conditions of adding coagulants to the lake water. Have the necessary permits been obtained from environmental protection agencies? Provide data in this regard.

Among the badly written parts are lines 133-156! Everything is in one sentence and very badly! It should be completely rewritten. For analyses performed in the laboratory, write the name of the laboratory and the country.

Explanations should be given about the cost of the proposed method and whether it is affordable or not, and the effect of the coagulants used on the lake ecosystem due to the possibility of contamination.

The discussion section is written without subsections and with long paragraphs that are boring! If possible, define subsections there and make paragraphs smaller.

Overall, I believe the manuscript needs substantial revision.

Comments on the Quality of English Language

A major concern with this manuscript is its writing style, as some parts are badly written with many sentences being long and boring! In this regard, the manuscript should be revised and rewritten.

Author Response

Dear Reviewer

We want to thank you very much for the review. We think your comments are very accurate and necessary to improve the quality of the manuscript. We made all your remarks in the text. We hope that the changes introduced by us following your recommendations will be sufficient for the manuscript to be accepted for publication.

  1. Throughout the manuscript, whenever possible, we have shortened sentences that are too long to avoid them being boring
  2. We changed the title of our manuscript according to your suggestion.
  3. At the end of the Introduction chapter, it is emphasized that the team of authors of the manuscript is the author of the innovative solution in the phosphorus inactivation method, and in manuscript, we describe the results of our work.
  4. In the description of the research object, we added that Lake Mielenko is located in Poland.
  5. We improved Figure 1 according to your suggestion.
  6. We have added information about climatic conditions and the use of the lake.
  7. In section 2.2 we added criteria according to which water samples were collected for testing.
  8. Additional information on coagulant applications and their environmental impact has been added in section 2.3. We added the cost of restoration.

During application, attempts were made to distribute the coagulants as evenly as possible over the entire surface of the designated water area and to strictly avoid uncontrolled discharges of the agent used during the standstill or maneuvers of auxiliary vessels. The coagulant was fed just below the water surface using technical solutions that prevented the formation of flocs from being aerated and their flotation.

The doses of coagulants were selected and applied in such a period (spring, autumn) that they did not cause any negative changes in water chemistry and at the same time effectively eliminated the mineral phosphorus content in the water. It should be clearly emphasized that all coagulant components are elements commonly found in nature.

The innovative method of phosphorus inactivation proposed by our team using two types of coagulants - aluminum and iron - is a very cheap technological solution. The cost of purchasing coagulants with the application is approximately EUR 15,000. The restoration of Lake Mielenko required water law consent, which was issued by the Polish Waters State Water Management Company based on a water law report.

  1. The form of writing in lines 133-156 has been changed to make the text legible. A place was added where analyses of collected water samples were performed.
  2. It is not possible to divide the Discussion section into subsections. Where possible, we shortened the sentences.

Reviewer 3 Report

Comments and Suggestions for Authors

The comments are on the text.

Comments on the Quality of English Language

The comments are on the text.

Author Response

Dear Reviewer,

We want to thank You very much for the review. We think that Your comments are very accurate and necessary to improve the quality of the manuscript. We made all your remarks in the text. We hope that the changes introduced to me following Your recommendations will be sufficient for the manuscript to be accepted for publication.

  1. To Abstract We added a sentence related to the Introduction: The process of accelerated eutrophication forces the search for innovative, effective methods to restore good quality of surface waters.
  2. Throughout the manuscript, we have clearly distinguished all forms of phosphorus, i.e. mineral P as Pmin., organic P as Porg. and total P as TP
  3. We have given in brackets that PAX is the trade name of the aluminum coagulant - polyaluminum chloride and that PIX is the trade name of the iron coagulant - iron chloride
  4. The use of iron salts in the coastal areas of the lake is safe for aquatic organisms because iron is an element commonly found in nature. Iron is not toxic in any oxidation state, in the absence of oxygen or overoxygenated conditions, as well as in a wide range of pH.
  5. We changed the sentence in line 80 (changed to of)
  6. Elongation is the ratio of Maximum length (m) to Maximum width (m). This parameter has no units.
  7. We improved Figure 2
  8. We removed double Hach in line 151
  9. In the section Material and Methods we have added a method for measuring EC and chlorophyll a
  10. We changed the sentence to: An alternative hypothesis tested was the presence of significant differences in mean annual values of tested parameters between 2018 (protective actions) and 2019 (after protective actions), 2020, 2021 (during restoration treatments), and 2022 (after restoration treatments).
  11. In 2020, we achieved an increase in the average organic phosphorus content compared to 2019 because of unusual weather conditions. The lake generally froze over in the winter of 2019/2020, which resulted in the intensification of the mineralization processes of organic matter. The mineral forms of nutrients released in this process were taken up by algae in the process of photosynthesis, which took place intensively throughout the year, and not as usual, i.e. from April to September. Thanks to the ongoing reclamation, this problem was at least limited. In the following years, the situation improved.
  12. We changed the word ammonia to ammonium in line 228
  13. We believe that the increase in ammonium nitrogen content in 2021 is related to the reduction of production processes due to the reduction in the amount of mineral phosphorus after the first stage of lake reclamation. Ammonium nitrogen is the mineral form of nitrogen most preferred by algae. Due to the improvement of oxygen conditions at the bottom of the lake, ammonification of organic compounds took place, and ammonium nitrogen was produced, the excess of which was not used by algae, which were becoming fewer and fewer.
  14. We changed was to were in Line 251
  15. We changed a short sentence in Line 289
  16. Based on research of the waters and bottom sediments of Lake Mielenko, as well as the hydrological and catchment situation, we found that the optimal solution is the phosphorus inactivation method. First of all, removing bottom sediments is a good method, but the sediments of the lake in question did not contain such high amounts of pollutants as to be suitable for removal. This is a very expensive method, so you need to have a good justification to use it. Secondly, there would be no space around the lake for storing and processing the extracted sediments. Covering the sediments is also not a good idea due to the high content of organic matter in the sediments of Lake Mielenko. Covering the sediments would cut off the supply of oxygen and, as a result, anaerobic decomposition processes would begin, releasing gases that would cause the insulation to lift. Biomanipulation is too ineffective to significantly improve the trophic state of the lake.
  17. The introduction of coagulants into lake water triggers reactions that lead to the binding of phosphates from the water column, both as a result of chemical reactions and adsorption on the resulting flocs of hydrolyzed coagulant. The coagulant flocs together with the bound by-products fall to the bottom, forming a thin coating on the sediment surface. For many years, this coating captures newly arriving phosphates from the lake environment and prevents the release of phosphates from bottom sediments. In Długie Lake in Olsztyn, Poland, we used the phosphorus inactivation method in 2001, 2002, and 2003. The mentioned coating still works today. This lake has crystal-clear water, and the release of phosphates from sediments is stopped even in the absence of oxygen at the bottom. Water transparency is on average 5m. The bottom is covered with macrophytes, the number of which was once limited by the lack of light. We will certainly observe such positive changes in Lake Mielenko in the coming years.

Round 2

Reviewer 1 Report

Comments and Suggestions for Authors

According to the authors' answers, it is also revising manuscript finally. It is accepted

Author Response

Dear Reviewer,

Thank you very much for the review and accept our corrections prepared according to your suggestion.

With kind regards

Authors

Reviewer 2 Report

Comments and Suggestions for Authors

Although the authors have made corrections, there are still some shortcomings. Comments are annotated on the attached PDF file.

Comments on the Quality of English Language

Moderate editing of English language required.

Author Response

Dear Reviewer,

We want to thank you very much for the again review of our manuscript.

We prepared the next version of our manuscript according to your new suggestion. We made all your remarks in the text. We hope that the changes introduced by us following Your recommendations will be sufficient for the manuscript to be accepted for publication.

  1. We cited the following references: Ye, X., Bu, W., Hu, X., Lin, B., Liang, K., Chen, F. Species divergence in seedling leaf traits and tree growth response to nitrogen and phosphorus additions in an evergreen broadleaved forest of subtropical China. Journal of Forestry Research 2023, 34, 137-150, doi.org/10.1007/s.11676-021-01437-2.
  2. We changed the last two paragraphs of the introduction section to “An innovative, sustainable solution developed by our scientific team is the restoration of lakes via modified P inactivation method. The research aims was implement of new method based on the sequential application of coagulants: iron chloride (trade name PIX 111) and polyaluminium chloride (trade name PAX 18) and to determine the effectiveness of this innovative method in the restoration of a shallow, polymictic urban lake. The iron coagulant was dosed in the coastal zone of the lake, where the water is well-oxygenated. The aluminum coagulant was dosed in the profundal zone, where the deep, over-bottom water is anoxic. This solution enables the reduction of restoration costs and ensures a higher level of ecological safety”.
  3. We added more detailed information about the climate in Kartuzy: “The climate of this region is cool and humid with a short growing season, which lasts 152 days, from May to October. In this period the average daily temperature exceeds 17 °C, which ensures the optimal development of plants and animals. In Kartuzy, the average annual air temperature is 6.5 °C. The hottest month of the year is July when the average daily temperature is 21°C, and the coldest is January, when the average daily temperature is -5 °C. The average annual rainfall of this region is 750 mm. The month with the highest rainfall is July - the average daily rainfall is 59 mm. The month with the least rainfall is February - the average daily rainfall is 13 mm. The period of the year with snowfall lasts 5 months - from November to April, and the average monthly snowfall during this time is 25 mm. The snowiest month is December, where the average monthly snowfall is 109 mm”.
  4. We changed Figure 1
  5. We have changed the description of water sampling: According to the guidelines used in limnological research, water samples for analysis should be collected at the deepest point of the lake from a depth of 1 m below the water surface and 1 m above the bottom. Due to the small depth of Lake Mielenko, samples were taken from a depth of 1 m, but at three points located at the vertices of a triangle with sides of 50 m in the deepest region of the lake. These points were located in the area of the maximum depth of the lake. During the research, a total of 180 water samples were taken for analysis. Water for analysis was collected using a Ruttner sampler (3.5 L, KC Denmark; Geomor Technik, Szczecin, Poland) into 2 L containers.
  6. Following your suggestion, we have changed the description of research methods based on the methods of their description contained in renowned scientific journals such as the Journal of Environmental Management, Water Research, Stoten, and Water
  7. We have added the following information in the section Restoration of Lake Mielenko: During the application of coagulant iron concentrations in the water of Lake Mielenko did not exceed 0.5 mg Fe/L (permissible concentration in water is 1 mg Fe/L), and aluminum concentrations during coagulant dosing were traces (permissible concentration in water is 0.3 mg Al/L).
  8. We have moved information about restoration costs and water law consent to the Discussion section.

With kind regards

Authors

Reviewer 3 Report

Comments and Suggestions for Authors

lines 344-345:

correct this sentence: "As research has shown [48] showed that phosphates have the greatest affinity for iron."

Comments on the Quality of English Language

lines 344-345:

correct this sentence: "As research has shown [48] showed that phosphates have the greatest affinity for iron."

Author Response

Dear Reviewer,

Thank you very much for the review and accept our corrections prepared according to your suggestion. We corrected the sentence As research has shown [48] showed that phosphates have the greatest affinity for iron."

With kind regards

Authors

Round 3

Reviewer 2 Report

Comments and Suggestions for Authors

The authors have made the desired corrections and I think the manuscript is acceptable.

Comments on the Quality of English Language

Minor editing of English language required.